# Intra-Individual and Intraspecific Terpenoid Diversity in *Erodium cicutarium*

**DOI:** 10.3390/plants10081574

**Published:** 2021-07-30

**Authors:** Elisabeth Johanna Eilers

**Affiliations:** Department of Chemical Ecology, Bielefeld University, Universitätsstraße 25, 33615 Bielefeld, Germany; elisabeth.eilers@uni-bielefeld.de; Tel.: +49-(0)-521-106-2597

**Keywords:** Geraniaceae, specialized metabolites, gas chromatography-mass spectrometry, terpenoid chemotypes

## Abstract

The chemodiversity between and within individuals of several plant species is remarkable and shaped by the local habitat environment and the genetic background. The forb *Erodium cicutarium* (Geraniaceae) is globally distributed and partly invasive. This paper hypothesizes a high intra-specific and inter-individual chemical diversity in this species and investigates this by comparing the concentration and diversity of terpenoid compounds in different plant parts, i.e., leaves, blossoms and fruits. Plants were grown from seeds, originating from native range Bavaria (BY), Germany, and invaded range California (CA), USA, populations. In total, 20 different terpenoids were found, which occurred in distinct combinations and the patterns clustered into groups of distinct chemotypes for all plant parts. Several of the chemotypes were specific to plants of one region. The terpenoid compositions of different plant parts within individuals were highly correlated. Chemodiversity was higher in reproductive plant parts compared to the leaves, and higher in plants from BY compared to CA. This study highlights the intra-specific and inter-individual chemodiversity in *E. cicutarium*, linked to its geographical origin, which may facilitate its invasion success but also calls for further investigation of the role of chemodiversity in invasive plants on interactions with the environment.

## 1. Introduction

Phytochemical diversity may shape species interactions and can be defined as a multidimensional concept that involves the complexity of phytochemical patterns and the variation thereof across spatial and temporal scales [1]. The composition of specialized metabolites of plants, which are stored in different organs and are in part emitted as volatiles, are remarkably different within several plant species [2]. Such chemical diversity does not only exist between plant families but also within one plant genus, species, or even within one individual plant (e.g., [3]). In general, increased trait diversity may allow for an increased potential for adaptations to new environments and thus an increased invasion success (e.g., [4]). Furthermore, abiotic and biotic environmental factors promote trait plasticity in plants at evolutionary timescales [5]. From an economically applied perspective for pharmaceutically used plants, it is particularly important to know the exact composition of the pharmaceutically relevant metabolites and the diversity within a plant individual and species. Increased phenotypic plasticity of invading plants compared to native plants are often linked to increased invasion success [6]. Whether increased plasticity in the chemical composition among and within invading plant species functions in a similar way is less known but this knowledge could help to better predict if a plant species may become invasive or not.

The production of phytotoxic metabolites and release as allelochemicals may help a plant to survive in highly competitive environments [7]. Plant-part specific differences in the amount and composition of specialized metabolites may relate to plant-part specific distribution of herbivores [8] and/or to the value of the respective tissue or organ [9]. Both, the presence of herbivores or other stressors and the value of a tissue, however, depend on the prevailing environmental conditions at the inhabited geographical area.

The annual forb *Erodium cicutarium* (L.) L’Hér., Hort. Kew. [W. Aiton] (Geraniaceae) is an interesting study species and has been investigated in former studies for several reasons. For instance, the shape and structure of *E. cicutarium* seeds and awns are unique and allow for ballistic dispersal and strong anchorage behavior. The plant is exotic, native to Europe, North Africa and Western Asia [10,11] and grows preferably in disturbed habitats [12]. The plant was probably accidentally taken from Europe to California in the 18th century and since then became a dominant invasive plant in large parts of North America [13,14]. To date, *E. cicutarium* is of economic importance as it causes yield losses in several crops: up to −92% in peas, −82% in dry beans, −37% in canola and −36% in wheat [15,16]. Aqueous *E. cicutarium* leaf extracts reduced radicle length of *Zea mays* and radicle and hypocotyl length of *E. cicutarium* seedlings [17].

Numerous studies report on high concentrations of specialized metabolites in *E. cicutarium*, such as terpenoids [17,18,19,20], flavonoids [21] and phenolic acids [22]. Terpenoids are among the most structurally diverse plant metabolites [23] and they are involved in a broad range of ecological functions, such as defense against herbivores and pathogens, allelopathy, nutrient cycling and attraction of pollinators [24]. Intraspecific terpenoid diversity can take on remarkable proportions within various plant families, e.g., herbaceous Asteraceae [25,26,27], Lamiaceae [28] or woody Fagaceae and Myrtaceae [29,30].

Interestingly, the different studies on the terpenoid composition of *E. cicutarium* plants came to strikingly different results, i.e., all studies found unique terpenoids, which were not described in the other studies [18,19,20]. The different findings may in part result from methodological differences. However, the investigated plants were field-collected from different populations at different geographical regions and thus exposed to different environmental growth conditions and different findings may indicate a high terpenoid diversity of *E. cicutarium* plants among different populations. Yet, we know little about the intraspecific and intra-individual, i.e., plant-part specific variability of specialized metabolites in *E. cicutarium*, or the function thereof. In a previous study on *E. cicutarium* we showed that the constitutive concentrations of foliar monoterpenes were twice as high in plants originating from populations that experienced high levels of competition compared to plants from low competition sites [17]. However, it remained unknown whether concentrations and diversity of terpenoids also differ in plants originating from different geographical areas. Although terpenoids are transported within plant individuals among different tissues [31,32], the ecological functions of vegetative and reproductive tissues of plant individuals obviously differ and consequently it can be speculated that differences in terpenoid diversity exist. However, intra-individual differences in terpenoid diversity remain largely unexplored to date.

The aim of this study was to characterize terpenoid profiles in plants originating from Bavarian populations (i.e., the native range) and from Californian populations (i.e., the invaded range) in order to shed light on apparent incoherence in the literature regarding terpenoids in *E. cicutarium*. Within each plant individual, terpenoids were solvent-extracted from three plant parts, i.e., leaves, blossoms and fruits.

Firstly, it was hypothesized to find intraspecific differences in terpenoid profiles that cluster into distinct chemotypes. Secondly, it was expected that the chemotypes of leaves, blossoms and fruits within plant individuals would be correlated because terpenoids of particularly high abundance in leaves would also be present in flowers and fruits of the same plants. In other plants such as *Tanacetum vulgare*, the terpenoids with the highest compound abundance, which shape the terpenoid chemotypes in leaves, are also found in high abundance in inflorescences [33,34]. Thirdly, it was hypothesized that the diversity of contained terpenoids would be high in plants from both, the native and the invaded range. Fourthly, despite overlaps of the main terpenoid components in *E. cicutarium* leaves, blossoms and fruits, it was expected that a higher number of compounds would be found in reproductive tissues, resulting in higher chemodiversity in blossoms and fruits due to the different functions of plant tissues. Although the number of investigated plants from different seed sources used in this study may not be sufficient to predict the role of chemodiversity for the plants’ invasion success, the results provide the first hint in this direction and call for future research on chemodiversity of invasive plants.

## 2. Results and Discussion

### 2.1. Terpenoids in E. cicutarium Plants of Different Geographical Origin

In leaf, blossom and fruit heptane extracts of *E. cicutarium* plants from two geographical regions, Bavaria and California, this study found 21 compounds, of which 16 were monoterpenes, four were sesquiterpenes and one compound could not be identified (Table 1). Most of the detected compounds were detected in individual plants within all three plant part groups. The sesquiterpene germacrene D was detected in highest relative proportions in most samples and was present in the majority of samples (Figure 1). Two of the monoterpenes, allo-ocimene and α-terpineol, were exclusive for the reproductive tissues, i.e., blossoms and fruits. One monoterpene, *p*-cymene, was exclusively found in fruit samples, and one monoterpene, α-thujene, was exclusively found in two blossom samples (Table 1).

As an adaptation to environmental conditions at the local sites, plants can adapt the composition of specialized metabolites in their tissues. This study found that the presence and abundance of terpenoid compounds were strongly associated to the geographic region of seed origin. For instance, the compound α-thujene was not present in plants originating from California, *p*-cymene was also very rare in samples from California and, if present, only detected in traces. On the other hand, α-terpinolene was very common in samples from plants originating from California but rare/low abundant in samples originating from Bavaria. Another study on greenhouse-grown *E. cicutarium* originating from the UK found 17 terpenoid compounds in foliar leaf extracts with hexane [20]. Furthermore, in *E. cicutarium* hydro-distillation extracts from air-dried plants of field collected *E. cicutarium* from Serbia, in one case 29 terpenoids from aerial plant parts without blossoms or flowers [18] and in the other case 28 terpenoids from entire above- and belowground parts were found [19]. Interestingly, between the two studies from Serbia using hydro-distillation, only eleven of the detected terpenoids match. Furthermore, only four of the compounds found in this study, linalool, *α*-terpineol, *β*-caryophyllene and germacrene D, were also found in one of the studies from Serbia [18]. In the other Serbian study, however, linalool, α-terpineol and germacrene D were only detected in traces and *β*-caryophyllene was not found [19]. Only six compounds found in this study were also found in the study using plants from the UK [20], while three of these shared compounds were also shared with both Serbian studies [18,19]. The comparisons of different geographical origins within this study and the comparisons to studies from other geographical origins suggest that *E. cicutarium* plants from different geographic origins differ remarkably in their qualitative terpenoid composition. Intraspecific differences of the qualitative terpenoid composition are also known from other plant species. In lodgepole pine (*Pinus contorta*, Pinaceae) for instance, constitutive levels of phloem terpenoids also vary across a geographical range, and these variances are potentially caused by diverging climatic conditions. Within the regions caused by the diverging climatic conditions, the likelihood of mountain pine beetle attack is different [35].

### 2.2. Ecological Relevance of Terpenoids in E. cicutarium

The results of this study show that plants of the same species, naturally occurring in the same geographical region, differ greatly in the composition of their terpenoids (Table 1 and Table 2), even when grown for several generations under standardized laboratory conditions. In most plants, a certain proportion of available terpenoids from the tissue is emitted as volatile signals, e.g., to repel herbivores or attract pollinators. In flowers, terpenoids are the most diverse and most commonly occurring class of volatiles [36]. Together with other volatiles and floral color, terpenoids are in animal-pollinated plants, often crucial to advertise rewards such as nectar and pollen and attract flower visitors from a distance [37]. However, *E. cicutarium* is self-fertile [12,38] and the flowering period lasts several months (e.g., April to November in Germany, own observations). Intriguingly, although *E. cicutarium* is supposedly not animal pollinated, the terpenoid diversity found in the blossoms is comparatively high and the measured compound abundances were comparable to concentrations found in fruits and leaves. Therefore, *E. cicutarium* is a very interesting species to study the role of pollinator-independent floral volatiles. In comparison, the constitutive foliar and floral terpenoid compositions of a highly pollinator-dependent plant with immense terpenoid diversity in plant tissues, *Tanacetum vulgare* (Asteraceae), are similar, leading to comparable chemotypes in flowers and leaves [33,34]. Because the attraction of beneficial organisms may not play a major role for the chemical composition of *E. cicutarium* plant parts, the detected compounds (mostly terpenoids) potentially function as herbivore repellent or act against competing plants. In *Tanacetum vulgare*, plant part-specific metabolome profiles could directly be linked to plant part-specific herbivore abundances, i.e., of aphids [3]. Herbivore exclusion experiments and laboratory assays in the invaded range of *E. cicutarium*, i.e., Arizona and California, USA, showed that *E. cicutarium* and its two sister species *Erodium texanum* and *E. botrys* were not impacted by herbivory [39] or, in moist areas, consumed by slugs [40]. Overall, the plant is, if at all, mostly attacked by herbivores such as vertebrates, which cause rapid and massive loss of tissue. In the native range, i.e., the UK, and in the invaded range, i.e., Mediterranean habitats in southwestern Australia, faeces of the European rabbit *Oryctolagus cuniculus* contained *E. cicutarium* plant material [41,42]. Thus, all plant parts may be simultaneously affected by herbivore attacks and a strong constitutive phytochemical defense of the whole plant is required. Furthermore, several terpenoids are phytotoxic and may thus act as allelochemicals, e.g., monoterpenes released by false rosemary (*Conradina canescens*, Lamiaceae) [43] and sesquiterpenes released by the invasive bitou bush (*Chrysanthemoides monilifera*, Asteraceae) [44]. Moreover, aqueous terpenoid-containing leaf extracts of *E. cicutarium* showed an allelopathic function [17]. However, we do not yet know whether the chemotype of a plant or its geographical origin affects its allelopathic potential.

### 2.3. Terpenoid Chemotypes and Plant Part Specific Combinations Therof in E. cicutarium

According to cluster analyses and silhouette plots, ten chemotypes were identified within leaf samples, four chemotypes within blossom samples and seven chemotypes within fruit samples (Figure 1). This clustering into distinct, plant part-specific chemotypes is in line with the first hypothesis. Similar findings have been made for other Geraniaceae, for instance *Pelargonium capitatum*, which is grown for commercial use of its essential oil. The species also expresses distinct chemotypes, which occur in different geographic regions [45]. In this study, the most abundant chemotypes in samples from plants with CA origin were leaf chemotype L3 and blossom chemotype B2 both with main compound α-terpinolene and fruit chemotype F2 with main compound germacrene D (Table 2, Figure 1 and Figure 2). In leaf samples of five BY and three CA plants, which originated in each case from the same site in BY or CA but from different source families (i.e., seed collection sites), no terpenoids were detected and these plants were assigned to chemotype L9. Two leaf chemotypes contained only one compound: leaf chemotype L1 exclusively contained α-terpinolene and leaf chemotype L4 germacrene D.

In line with the second hypothesis, the chemotype of one plant part, e.g., the leaves, was linked to distinct chemotype- combinations of the remaining two investigated plant parts, e.g., blossoms and fruits (Figure 2 and Figure 3b). The blossom chemotype B1 was most commonly found in plants originating from BY populations (51% of plants). It occurred in BY populations in only three combinations with leaf and fruit chemotypes, of which the most common combination (61%) was leaf chemotype L5 and fruit chemotype F4. This combination of plant part-specific chemotypes occurred in only one of the plants originating from CA populations. In plants originating from CA populations, the blossom chemotype B1 was most frequently linked to leaf chemotype L2 and fruit chemotype F2 (57%), a combination which was, in turn, exclusive for plants from CA populations and not found in any of the plants originating from BY populations. The blossom chemotype B2 was most common (68%) in plants originating from CA populations, but not found in any of the plants originating from BY populations (Table 2). This blossom chemotype occurred in CA populations in only four combinations with leaf and fruit chemotypes, of which the most common combination (47%) was with leaf chemotype L3 and fruit chemotype F2 (Figure 2).

Tissues within individual plants contain characteristic metabolites and thus chemical communication displays, but the potential for independent variation of specialized metabolites within different tissues of the same plant individual is often limited by biosynthetic constraints and pleiotropic effects [46]. The high compound diversity found in *E. cicutarium* plants within this and in comparison to other studies is likely caused by different expression of terpene synthases, which are the primary enzymes that catalyze the synthesis of hemiterpenes (C5), monoterpenes (C10), sesquiterpenes (C15) or diterpenes (C20) [23,24]. Various terpene synthases catalyze the production of multiple terpenoids and biosynthetic intermediates from one substrate-type, thus producing complex product mixtures [23]. Monoterpene synthases from grand fir (*Abies grandis*) for instance, convert geranyl diphosphate to the monoterpenes myrcene, (-)-limonene, *β*-phellandrene, (-)-α-pinene and (-)-*β*-pinene [47]. The fact that one enzyme can provoke the conversion into a specific group of terpenoids likely explains the co-occurrence of compounds, and thus correlations of compound abundances between plant parts (Figure 2 and Figure 3b). Future studies are required to investigate the biochemical background of chemotype formation in *E. cicutarium*.

### 2.4. Intra-Individual and Intraspecific Terpenoid Diversity in E. cicutarium

In blossom and leaf samples, the compound diversity was significantly higher in samples from plants originating from seeds collected in Bavaria (BY), i.e., within the native range, compared to samples from plants with Californian (CA) origin, i.e., within the invaded range (Figure 4; Χ 2 leaves = 5.17, *p* = 0.023, *n* = 159; Χ 2 blossoms = 4.06, *p* = 0.044, *n* = 144; Χ 2 fruits = 0.15, *p* = 0.7, *n* = 114). The significant difference in the terpenoid diversity between plant samples from Bavaria (BY, native range) compared to samples from California (CA, invaded range) are in contradiction to the third hypothesis. Similar to the finding for chemodiversity, the average number of compounds contained in samples was highest for fruits, followed by blossom and leaf samples (mean ± s.d. leaves: 3.82 ± 2.51; mean ± s.d. blossoms: 9.09 ± 3.56 mean ± s.d. fruits: 13.41 ± 2.59). There were, however, no significant differences in compound numbers between both regions (Χ 2 leaves = 2.14, *p* = 0.143, *n* = 159; Χ 2 blossoms = 0.47, *p* = 0.494, *n* = 144; Χ 2 fruits = 0.04, *p* = 0.833, *n* = 114). This suggests that the evenness of the relative terpenoid composition is lower in CA plants, or in other words, CA plants contain on average a higher proportion of dominant compounds. One reason for a lower terpenoid diversity from plants in the invaded range may be a genetic bottleneck caused by the invasion process or post-introduction evolutionary processes. Whether such a genetic bottleneck occurred in Californian *E. cicutarium* plants remains speculative, and further studies are essential to gain further clarity. Wolf et al. [48] proposed that an increased chemical diversity could facilitate the invasion success of a plant species. However, the results found in this study are not sufficient to supporting the chemical diversity-invasion hypothesis and further experiments, i.e., common garden experiments with plants of varying terpenoid diversity at both ranges are required. Another possible explanation for a lower terpenoid diversity from plants in the invaded range would be differences in abiotic environmental factors between both geographical origins. In this study, the main difference in abiotic environmental factors between the German and Californian sites is the average temperature. Therefore, it is necessary to obtain further samples from the native and invaded range of *E. circutarium* from geographical areas with comparable climates to gain further clarity in this regard. Finally, specialized metabolites differ in many invasive plants due to the escape from specialist herbivores in the invaded range [49]. However, little is known about the release of enemies in the invaded range of *Erodium cicutarium*.

According to random forest models, the patterns of compound abundances did not separate into distinct clusters of plant parts or seed origins (Figure 3a). However, a trend for cluster separation was present, particularly with regard to the two geographical regions of plant origin. The most important compounds that explain the trend in random forest cluster separation were α- terpinolene for differences of compound patterns based on the region of plant origin (Figure 3c) and germacrene D for differences of compound patterns based on the plant part (Figure 3d). Mantel tests between the compound patterns of samples from the three plant parts revealed significant correlations between the three plant part groups (Figure 3b).

The compound diversity, measured as Shannon diversity index of relative compound abundance, was higher in blossom and fruit samples compared to leaves (mean ± s.d. leaves: 0.87 ± 0.57; mean ± s.d. blossoms: 1.71 ± 0.45 mean ± s.d. fruits: 1.62 ± 0.28) (Figure 4). The difference in terpenoid diversity between the three investigated plant tissues is in line with the fourth hypothesis. The broad range of ecological functions of terpenoids within different plant organs, such as allelopathy, nutrient cycling, attraction of mutualists and defense against antagonists [24], indicates that various abiotic and biotic environmental factors may cause adaptations in constitutive terpenoid contents of plants to their environments.

## 3. Materials and Methods

### 3.1. Origin of Plant Seeds

Seeds of *E. cicutarium* were collected in 2011 and 2012 from four populations in two geographical regions: Bavaria, Germany (continental climate, native range) and California (USA, Mediterranean climate, invaded range) (for further details on the seed collection procedure, see [50]). Of these seeds, five maternal families (i.e., mother plants) per population were included in this study. During sampling, the percentage of surrounding plant cover was estimated and aboveground biomass by neighbor plants of the *E. cicutarium* plants was measured at the field collection sites in three 50 × 50 cm plots. Within each region, two populations with a comparably high level of competition (defined as >73% cover of grasses and other plant neighbors) and two populations with a comparably low level of competition (defined as <35% cover) were chosen (see [17,50]). The abiotic environmental conditions at the sample sites of the different populations were as similar as possible. Heger and colleagues measured and statistically compared soil nutrient levels (Ca++, K+, Mg++, NH4+, NO2−, NO3−, PO4−−−, SO4—) and the percentage cover with surrounding vegetation and both factors showed no significant difference between the regions [50]. The mean annual precipitation and mean annual temperature were 798 ± 28 mm (s.e.) and 8.46 ± 0.16 °C (s.e.) in Germany and 824 ± 78 mm (s.e.) and 14.71 ± 0.43 °C (s.e.) in California. The second generation of offspring, grown under standardized conditions in a greenhouse, served as experimental plants in this study. The same plants were used to investigate the effects of former plant competition on aboveground biomass, fertility, seed traits and total foliar terpenoid concentration on offspring plants in another study [17].

### 3.2. Cultivation and Harvest of Plants

Eight seeds per source family (40–80 per population) were placed for germination on 1.2% agar-agar (Bioscience, Roth, Germany) in transparent plastic boxes with semi-transparent lids (l × h × w = 200 × 65 × 200 mm). The boxes were placed in a climate chamber at 20 °C and 70% humidity 16:8 h light: dark. After 7 days, seedlings were transferred into substrate (2:1 river sand: garden mold) in individual pots (l × h × w = 7 × 8 × 7 cm, polypropylene) and grown under the same conditions. Differences in germination rates [17], resulted in unequal sample sizes of experimental plants for the two geographical areas and source families. The plants were watered thrice a week and additionally fertilized twice a week with a standard formulation of mineral fertilizer (Wuxal, Manna NPK fertilizer solution 8-8-6 with trace nutrients, Düsseldorf, Germany), starting two weeks after transfer into pots. The fertilizer contained 8% total nitrogen, 8% diphosphorus pentoxide and 6% potassium oxide.

From plants which had already produced blossoms and unripe (soft and green) schizocarps, including the calyx 16 weeks after placing the seeds on agar (i.e., day 112), leaves and reproductive parts were harvested. Schizocarps are fruits containing up to five diaspores consisting of a seed and a connected awn. The five largest leaves of each plant were combined to respective leaf samples, all open flowers (complete, including sepals and receptacles) and all flower buds of plant individuals were combined to respective blossom samples, and a randomized sample of 5–7 unripe schizocarps of each plant individual was combined to respective fruit samples. The plants which did not flower after 16 weeks (approx. 15%) were harvested two weeks later in order to collect tissues from plants of comparable ontogenetic state. All plants had produced at least the first flower buds when harvested. However, the dry weight of blossoms and/or unripe fruits was not in all cases sufficient for chemical analysis, resulting in *n* = 159 samples for leaves, *n*= 144 samples for blossoms, *n* = 114 samples for fruits (i.e., unripe schizocarps). Directly after harvest, the samples were placed in liquid nitrogen, subsequently frozen at −80 °C and lyophilized for 24 h before terpenoid extraction.

### 3.3. Terpenoid Extraction and Analysis of Leaves, Blossoms and Fruits

For extraction and analysis of terpenoids, a modified protocol of [51] was used. First, the lyophilized plant material was coarsely crushed and 30–40 mg of leaf 15–25 mg of blossom and 15–25 mg of unripe fruit samples was weighted. Thereby, the exact amount for each sample was noted. A stainless-steel ball (Ø 5 mm) and 1 mL n-heptane (Roth, 99% HPLC grade), containing 100 ng/µL 1-bromodecane (97%, Sigma Aldrich, Karlsruhe, Germany) as internal standard were added to each sample and the samples were ground in the solvent for 30 s at 30 Hz (Retsch MM 301, Haan, Germany). Then, all samples were incubated for 15 min in an ultrasonic bath at room temperature and centrifuged for 5 min at 13,200 rpm. Before the samples were analyzed by gas chromatography coupled with mass spectroscopy (GC-MS), the supernatants were concentrated to a volume of approximately 40 µL. The GC-MS column was a VF-5 ms column (30 m × 0.2 mm ID, 10 m guard column, Varian, Palo Alto, CA, USA) and the electron impact ionization mode was used. Due to logistic reasons, leaf and blossom samples were measured on a Focus GC and DSQII (Thermo Electron, Rodano, Italy) instrument, while the fruit samples had to be measured on a GC 2010 PlusAF, MS QP2020 instrument (Shimadzu, Kyoto, Japan), in the same laboratory and using the same GC-MS parameters. Helium served as a carrier gas (flow rate = 1.1 mL/min). The oven program started at 50 °C for 5 min and increased at 280 °C with 10 °C/min. The MS scanned at 12.35 scans/s with a mass range of 35–400 amu. An alkane standard mix (C8-C20, Sigma Aldrich, Karlsruhe, Germany) was analyzed under the same conditions as the plant samples in this study to calculate retention indices [52]. By comparing the ms spectra of the samples to those of synthetic reference compounds (Sigma-Aldrich, Munich, Germany), where available, with proposals of the National Institute of Standards and Technology library as well as to retention indices and mass spectra published in the Pherobase [53], the compound identities were determined. In order to calculate relative terpenoid concentrations, the sample peak areas were normalized to the internal standard peak area of the same sample and afterwards the obtained peak areas were converted to the sample dry weight.

### 3.4. Statistical Analyses

For all statistical analyses, the program R, version 3.6.0 [54] was used and the statistical comparisons were based on the relative abundance of compounds detected in *E. cicutarium* samples. In order to determine the chemical diversity for each plant part of every plant, the Shannon index was calculated (Ref. [55]; function “diversity” in R package “vegan” [56]). The Shannon index is defined as H = −sum pi log(b) pi, where pi is the proportional abundance of a terpenoid i and b is the base of the logarithm. The Shannon index was then compared within plant parts among plants of the two geographic regions (Bavaria, native range and California, invasive range) by applying linear mixed models (LMM) with Gaussian distribution and a nested design, i.e., source family nested in source population, using the function “lmer” in package “lme4” [57]. The aboveground competition level to which ancestral plants in the field were exposed to (high, i.e., >73% or low, i.e., <35% cover of surrounding vegetation) was included as random factor in the models. In order to assign the investigated plants to different chemotypes, a hierarchical cluster analysis was applied with correlation distances applying the Ward method with 10,000 bootstrap replications on the whole dataset (methods “Ward.D2” and pyclust, Ward Jr., 1963 [58]). Silhouette plots were generated (package “cluster”, ref. [59]) to determine the number of clusters (= chemotypes) within each plant part. Only if data for all three plant parts were available, they were included in the cluster analyses (*n* = 99 plants). In order to visualize the co-occurrence of leaf, blossom and fruit chemotypes resulting from the cluster analysis, the R package bipartite was used through the function plotweb2 [60]. The default settings of the bipartite package version 2.16 were applied.

For comparisons of terpenoid profiles between the plant parts (i.e., leaves, blossoms, fruits) with random forest (RF) models (packages “randomForest” and “party”, [61]) and Mantel tests (package “vegan”, ref. [56]), the same data as for the determination of chemotypes (*n* = 99 plants) were used. For each RF classification tree (the number of RF trees = iterations were set to 10,000), five randomly selected variables were accepted as candidates at each split (mtry was set to 5 = approx. square root of the number of variables, i.e., the 21 compounds). From this unsupervised RF model, the results were displayed in multi-dimensional scaling of proximity matrix (MDS) plots. In supervised RF models for the two regions and for the three plant parts, the mean decrease in accuracy (MDA) was calculated, for the assumption that the respective variable was removed from the calculation in order to display the importance of each variable (i.e., chemical compound) in predicting the separation of data points. To compare the similarity of chemical compound patterns found in the three plant parts, the three datasets were transformed by Wisconsin double standardization of the square root values of the data, using the vegan package. For the Mantel tests, pairwise Kulczynski distances were used, 10,000 permutations were performed, and Spearman rank correlations were applied to compute Mantel’s r and *p*-values.

## 4. Conclusions

To the author’s knowledge, this is the first study that shows intra-individual and intraspecific differences in terpenoid profiles of *Erodium cicutarium* plants from different geographical origins by using the same analytical procedure. The different terpenoid patterns cluster in distinct chemotypes, which are unique for all three investigated parts of this plant. Comparisons of terpenoids in *E. cicutarium* found in this and other studies show a great variability among different geographical regions and the ecological consequences and evolutionary reason for the maintenance of such a high metabolic diversity remains unclear. In this study, the intra-individual terpenoid diversity was highest in blossoms. Hence, the self-fertile plant *E. cicutarium* is a very interesting model to study the role of pollinator-independent floral compounds, i.e., terpenoids and the diversity thereof. Furthermore, future studies are required to investigate whether terpenoids in *E. cicutarium* and their diversity are involved in anti-herbivore defense and/or how they may function as allelochemicals at different locations. Finally, the role of chemical diversity and chemical plasticity for the invasive success of this plant species remains to be elucidated in future studies.

## Figures and Tables

**Figure 1 plants-10-01574-f001:**
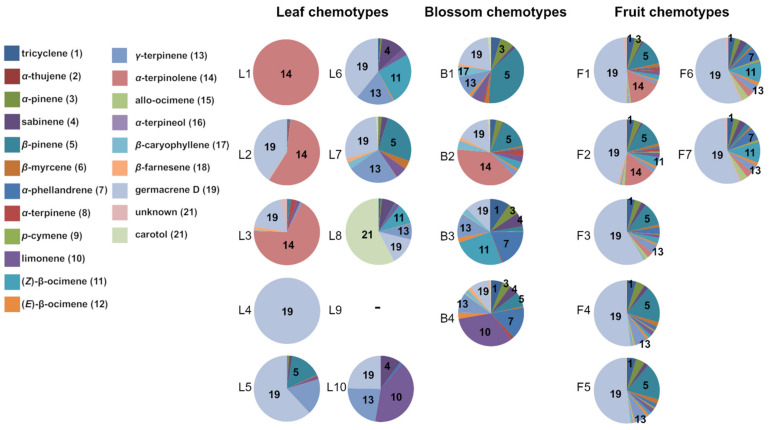
Relative compound abundance within each plant part-specific chemotype of *Erodium cicutarium* plants. Chemotypes were determined by hierarchical cluster analyses, using squared Euclidian distances and Ward’s method for linkage and by subsequently generating silhouette plots. Numbers in pie charts refer to numbers in brackets behind the compound names in the legend.

**Figure 2 plants-10-01574-f002:**
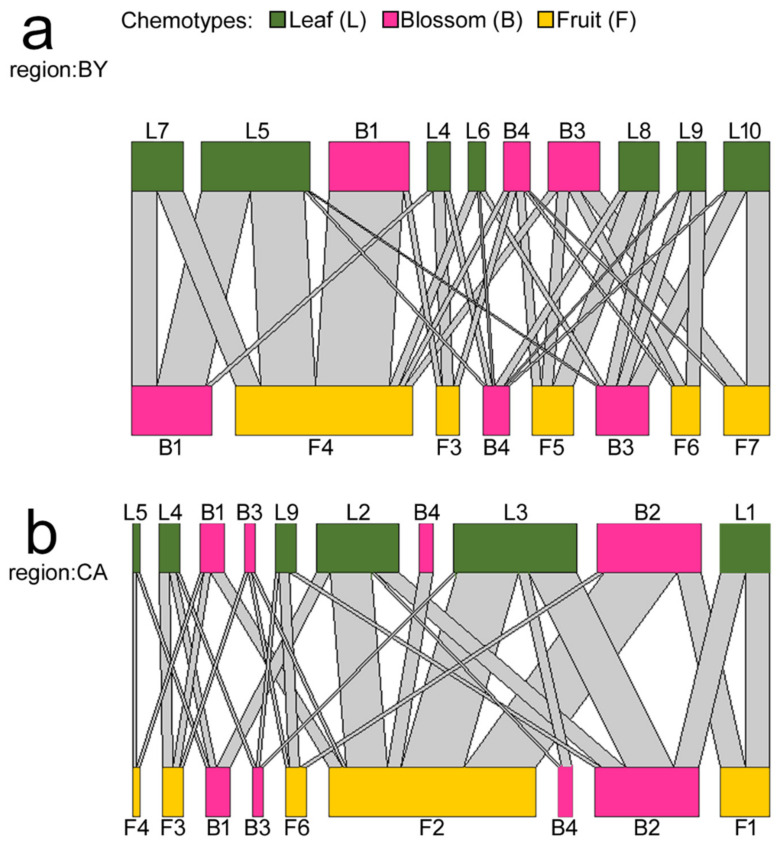
Comparison of the co-occurrence of leaf (L), blossom (B) and fruit (F) chemotypes for *E. cicutarium* plants originating from Bavaria ((**a**); BY, native range) and California ((**b**); CA, invaded range). The color of the nodes represents the plant organ (see legend). The width of the connections linking the nodes represents the number of co-occurrences. Several plant part specific chemotypes were only found in one of both regions (see Table 2). Please note that several organ-specific chemotypes are displayed twice in the Figure, as the triangular connection between flowers, leaves and fruits could not be displayed in a hierarchical tripartite plot.

**Figure 3 plants-10-01574-f003:**
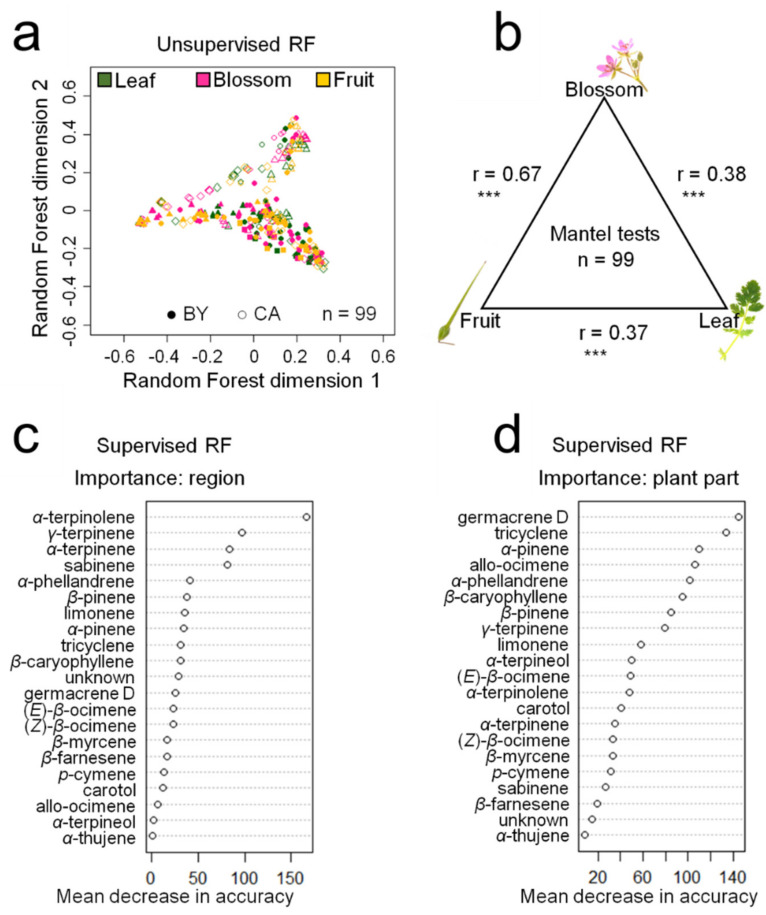
Random forest (RF) and Mantel statistic comparison of terpenoid profiles measured in different plant parts of *Erodium cicutarium* plants. Seeds originated from three Bavarian (native range) and three Californian (invasive range) populations (total *n* = 99 replicates per plant part). RF analyses results are displayed as point charts (**a**) with different symbols for the three different collection sites within each geographical region. The r-values and significance levels (*** *p* ≤ 0.001) of the Mantel tests are displayed in (**b**). Variable importance values are displayed as mean decrease in accuracy when removing the respective compounds from the analyses in supervised models using the factor “region” (**c**) or “plant part” (**d**) as explanatory variables.

**Figure 4 plants-10-01574-f004:**
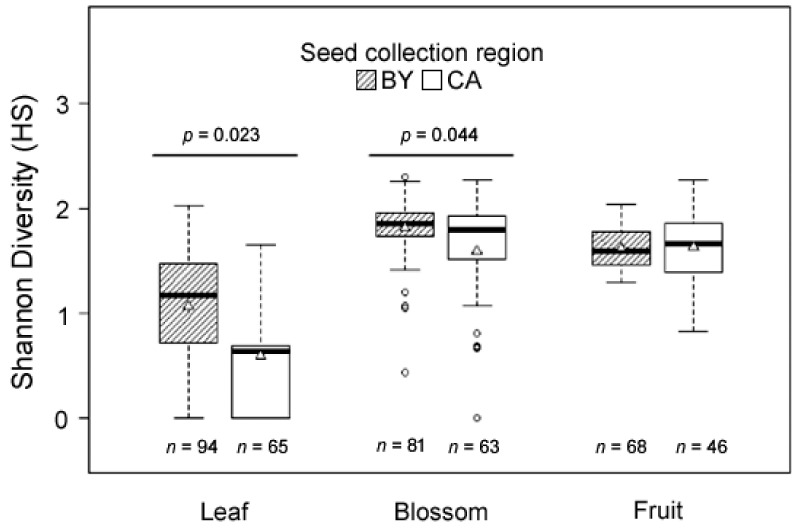
Shannon diversity index of relative terpenoid abundance detected in *Erodium cicutarium* plant parts. Seeds were collected in the field from Bavarian sites (BY, native range) and Californian sites (CA, invasive range). Plant parts of the second greenhouse generation were analyzed. The box plots represent medians (horizontal lines within boxes), upper and lower quartiles (box dimensions), 75% and 25% quartiles (whiskers) and outliers (circles). The mean values are shown by white triangles in boxes.

**Table 1 plants-10-01574-t001:** Identification of volatile compounds (monoterpenes (M) and sesquiterpenes (S)) detected in heptane extracts of leaves and blossoms (merged samples of open flowers and flower buds) of *Erodium cicutarium* plants. The mean content is the average of the percentage of the peak area for each compound in relation to the sum of the peak areas in the same samples and is given together with the standard deviation (s.d.). All compounds were identified by comparison of Kovats retention indices (*R*t) and comparison of mass spectra with NIST library data (mass spectrum match >95% and KI ± 3) or authentic standard compounds (*).

				Mean Content (%) ± s.d.			Detected in % of Samples				
#	R*t*	Name		Leaf	Blossom	Fruit	Leaf	Blossom	Fruit	BY	CA
1	930	tricyclene	M	0.10 ± 0.56	4.99 ± 4.50	4.26 ± 1.95	3	74	97	58	47
2	932	*α*-thujene	M	0	0.03 ± 0.21	0	0	2	0	1	0
3	941	*α*-pinene *	M	0.33 ± 1.14	5.27 ± 3.62	3.75 ± 1.30	11	71	97	65	52
4	976	Sabinene *	M	1.89 ± 4.27	2.36 ± 3.39	2.35 ± 2.55	23	56	89	68	31
5	986	*β*-pinene *	M	8.47 ± 11.63	22.34 ± 17.57	12.22 ± 9.11	42	76	87	66	65
6	990	*β*-myrcene	M	0.65 ± 1.96	1.10 ± 1.59	1.60 ± 0.91	11	40	85	47	33
7	1012	*α*-phellandrene	M	0.22 ± 1.23	5.48 ± 10.87	3.14 ± 5.54	4	41	50	28	32
8	1024	*α*-terpinene	M	0.41 ± 1.46	1.51 ± 2.94	0.60 ± 0.75	8	44	61	23	51
9	1031	*p*-cymene	M	0	0	1.11 ± 3.87	0	0	23	9	3
10	1037	Limonene *	M	4.26 ± 9.52	6.09 ± 9.37	1.77 ± 1.14	36	66	77	65	48
11	1040	(*Z*)-*β*-ocimene	M	0.92 ± 4.14	5.26 ± 10.18	4.13 ± 6.32	6	39	75	42	29
12	1050	(*E*)-*β*-ocimene	M	0.08 ± 0.34	1.42 ± 1.83	1.28 ± 0.81	5	48	84	48	33
13	1065	*γ*-terpinene	M	10.25 ± 11.56	7.83 ± 6.21	3.97 ± 2.53	54	84	88	92	49
14	1090	*α*-terpinolene	M	25.24 ± 34.98	13.32 ± 21.52	5.94 ± 8.34	36	32	44	4	82
15	1131	allo-ocimene	M	0	0.14 ± 0.51	1.44 ± 1.49	0	8	92	29	26
16	1206	*α*-terpineol	M	0	0.02 ± 0.16	0.29 ± 0.38	0	3	49	14	14
17	1447	*β*-caryophyllene *	S	0.54 ± 1.63	4.76 ± 5.22	0.65 ± 0.59	11	69	64	46	45
18	1456	*β*-farnesene	S	0.78 ± 2.54	0.93 ± 2.08	0.44 ± 1.30	11	33	42	26	29
19	1506	germacrene D *	S	35.00 ± 25.63	14.76 ± 9.35	50.39 ± 13.11	84	90	100	94	86
20	1523	unknown	S?	0.33 ± 1.58	0.21 ± 1.02	0.58 ± 1.49	4	7	16	5	14
21	1638	carotol	S	4.24 ± 17.32	0.81 ± 1.36	0.10 ± 0.15	7	33	30	18	29

**Table 2 plants-10-01574-t002:** Occurrence of plant-part specific chemotypes (for chemical composition see Figure 1) in *Erodium cicutarium* plants. The number of plants, which were found to belong to the different leaf, blossom and fruit chemotypes are listed for the two geographic regions of plant origin: Bavaria (native range) and California (invasive range). Total *n* = 99 plants (55 from Bavaria and 44 from California).

		Geographical Region	
Plant Part	Chemotype No.	Bavaria	California
Leaf	L1	-	7
Leaf	L2	-	12
Leaf	L3	-	18
Leaf	L4	4	3
Leaf	L5	19	1
Leaf	L6	3	-
Leaf	L7	9	-
Leaf	L8	7	-
Leaf	L9	5	3
Leaf	L10	8	-
Blossom	B1	28	7
Blossom	B2	-	30
Blossom	B3	18	3
Blossom	B4	9	4
Fruit	F1	-	7
Fruit	F2	-	30
Fruit	F3	4	3
Fruit	F4	31	1
Fruit	F5	7	-
Fruit	F6	5	3
Fruit	F7	8	-

## Data Availability

The data is uploaded to the Knowledge Network for Biocomplexity (KNB).

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
