# Peer review of "Intra-Individual and Intraspecific Terpenoid Diversity in Erodium cicutarium"

_plants, 2021, doi:10.3390/plants10081574_

Round 1

Reviewer 1 Report

The manuscript aims to study the intra-individual and intraspecific terpenoid diversity in Erodium cicutarium. In the current form, the text is hard to follow and needs rearrangement. First of all, I strongly recommend rewriting the text in passive voice. The novelty of the study is not emphasized. Moreover, there is no clear statement why it is important to study E. cicutarium and its intra-individual and intraspecific terpenoids diversity. From the literature, it is already seen that differences are expected.

More of my critical questions and comments are listed below.

Why did the author decide to collect seeds and place for germination in laboratory conditions rather than collect the leaves, blossoms, and flowers from the field and analyze them directly? The author states that the abiotic and biotic environmental factors promote trait plasticity in plants at evolutionary timescales. When changing these conditions planting the seeds in laboratory conditions doesn’t you provoke further changes in metabolite composition? In this regard, how representative is your design and results when no direct comparison with the mother populations is done?

Abstract

In my opinion, the abstract needs to be reworded. 

A clear statement that seeds from two geographical regions were collected must be included.

L 10: Former studies on E. cicutarium from different geographical areas and on different plant parts describe very different terpenoids. – could be subtracted.

L 16-17: more concrete data must be cited.

Introduction

Nearly 60% of all references cited in this part are over ten years old. In my opinion, this is not a good attestation for the actuality of the topic.

L 40-45: It is an unnecessary paragraph; I suggest deleting it.

The importance of terpenoids and the need for their analysis is not well emphasized. I suggest having a paragraph before L 54-64.

I suggest separating the hypothesis from the aim in different paragraphs.

Results and Discussion

Table 1 must be placed after the first paragraph. As all figures should follow the paragraph where they were first mentioned.

L 178: In Table 1 do you present the result ±standard deviation or standard error?

А Тable for the terpenoid composition of the two populations (separately) would contribute to a better understanding of the manuscript.

Materials and Methods

L 399 : The exact composition of the mineral fertilizer must be included as on the label.

L 381-382: Between the German and the Californian sites, soil nutrient levels and the percentage cover with surrounding vegetation were comparable. Please, provide information about the soil nutrient levels and composition.

L 417: if the method of extraction and analysis is the same as your previous paper [17], I suggest referencing it and delete the description in this paper.

The conclusion is superficial and not appropriate, please reword it.

Although I opt for Major revision, this manuscript is more suitable for Short Communication than an original research manuscript.

Author Response

Dear Reviewer,

thank you for the detailed and constructive comments on my work. In the following, I will comment on the individual points (in blue):

The manuscript aims to study the intra-individual and intraspecific terpenoid diversity in Erodium cicutarium. In the current form, the text is hard to follow and needs rearrangement. First of all, I strongly recommend rewriting the text in passive voice. The novelty of the study is not emphasized. Moreover, there is no clear statement why it is important to study E. cicutarium and its intra-individual and intraspecific terpenoids diversity. From the literature, it is already seen that differences are expected.

It is a pity that you consider the text as “hard to follow” and I hope that the present changes have contributed to promote understanding. However, I did not follow the request to rewrite the text in passive voice for the following reasons: Most scientific journals (e.g. Science and Nature) encourage active voice because active-voice sentences are straightforward, thus clearer often more concise than passive-voice sentences. I share your opinion: chemical diversity has been reported in a wide array of studies in the past. Nevertheless, detailed and comprehensive knowledge is still missing and thus, this field remains highly exciting. For instance, the present work makes a very important contribution to the DFG-funded Research Unit FOR 3000 investigating the "Ecology and Evolution of Intraspecific Chemodiversity of Plants", established in 2020, in which 10 PIs and numerous PhD students from various renowned institutes in the field of Chemical Ecology are involved. In this research group, we are not only roughly describing intraspecific chemical diversity, as we know it from many older studies, but we aim to decipher the details, e.g. to which extent chemodiversity differs in the tissues within an organism and to which extent chemodiversity is determined by the geographical origin of the plants. These considerations are of high novelty and importance in order to understand what the ecological consequences of intraspecific plant chemodiversity are and how plant chemodiversity is genetically determined and maintained. I have emphasized the novelty, importance and specialty of this study more clearly at several points in the introduction now.    

More of my critical questions and comments are listed below.

Why did the author decide to collect seeds and place for germination in laboratory conditions rather than collect the leaves, blossoms, and flowers from the field and analyze them directly? The author states that the abiotic and biotic environmental factors promote trait plasticity in plants at evolutionary timescales. When changing these conditions planting the seeds in laboratory conditions doesn’t you provoke further changes in metabolite composition? In this regard, how representative is your design and results when no direct comparison with the mother populations is done?

Although the question how abiotic and biotic environmental factors promote trait plasticity is definitely very interesting, this was not my aim. I would need a much larger sample size to answer this question, especially to be able to disentangle which proportions of the chemical diversity arise from environmental factors and which parts are genetically determined. The terpene chemotype of a plant describes the constitutive levels of these compounds and this proportion is genetically determined. In this study, my aim is the characterization of those chemotypes. In other words, I am currently not interested in the short-term chemical adaptation of plants to their current environment (i.e. the combination of constitutive and induced compounds). My goal is it to show how great the chemodiversity in this plant is, even when it is not differentially exposed to UV, attacked by herbivores, and so on. How large the variation must be in the field is undoubtedly tremendous.

Abstract

In my opinion, the abstract needs to be reworded. 

A clear statement that seeds from two geographical regions were collected must be included.

Done (line 13 in revised ms). 

L 10: Former studies on E. cicutarium from different geographical areas and on different plant parts describe very different terpenoids. – could be subtracted.

Done.

L 16-17: more concrete data must be cited.

By “cited”, you mean listing own results, not citing literature, right? If this was meant, it is now implemented.

Introduction

Nearly 60% of all references cited in this part are over ten years old. In my opinion, this is not a good attestation for the actuality of the topic.

Yes, it is true that about half of the references mentioned in the introduction are older than 10 years. The literature sources were primarily selected according to their relevance, not just their age, and this resulted in a wide range of citations from 1983-2021 in the introduction (see Figure in attached file). However, I would interpret this as a strength of this study, as it uses new approaches and new ways of statistical data analysis to answer questions that scientists have been asking for a long time, but which have still not been answered satisfactorily.

L 40-45: It is an unnecessary paragraph; I suggest deleting it.

This paragraph clarifies why it is interesting to study metabolites in different plant parts from different geographical origins. If I delete the paragraph, this link from the theoretical background to my present work is missing.

The importance of terpenoids and the need for their analysis is not well emphasized. I suggest having a paragraph before L 54-64.

The missing information “Terpenoids are among the most structurally diverse plant metabolites [..]” was now shifted before the former lines 54-64.

I suggest separating the hypothesis from the aim in different paragraphs.

Done.

Results and Discussion

Table 1 must be placed after the first paragraph. As all figures should follow the paragraph where they were first mentioned.

Thank you for the advice, I have revised the order of the tables and figures now.

L 178: In Table 1 do you present the result ±standard deviation or standard error?

It is the standard deviation, as indicated by “s.d.” in the Table header. I added this information now additionally to the Table caption. 

А Тable for the terpenoid composition of the two populations (separately) would contribute to a better understanding of the manuscript.

Perhaps I do not fully understand what is supposed to be in the requested additional Table. By population here, do you mean the region (BY versus CA?), which includes multiple populations? The terpenoid composition of the plants of the different regions was now added to Table 1 and the chemotype abundance of both regions is compared in Table 3.

Materials and Methods

L 399 : The exact composition of the mineral fertilizer must be included as on the label.

Done.

L 381-382: Between the German and the Californian sites, soil nutrient levels and the percentage cover with surrounding vegetation were comparable. Please, provide information about the soil nutrient levels and composition.

I did not measure the soil nutrient data myself, but I received this information from Tina Heger [50], so I cannot insert the information here. However, I have now clearly pointed out which parameters were measured.

L 417: if the method of extraction and analysis is the same as your previous paper [17], I suggest referencing it and delete the description in this paper.

I am convinced that this information is important and should not be deleted here. In a previous version of this manuscript, submitted to another journal, I had described the methods more briefly and referred to the literature as suggested. Both of two reviewers at that time criticized exactly this point and asked me to include this information in detail.

The conclusion is superficial and not appropriate, please reword it.

Done.

Although I opt for Major revision, this manuscript is more suitable for Short Communication than an original research manuscript.

Thank you for stating your opinion.

Reviewer 2 Report

Manuscript is interesting and quite well written. I have only some comments regarding the organization:

  • Tables and Figures should be placed close to the point where they were cited in the text to facilitate following the content. Figures should be cited first and then placed in the text (comment for Fig. 2).
  • Line 255: Fig. 3 should be cited before Fig. 4
  • Line 373: Why was reference [50] placed here?
  • The scientific text are rather written in passive voice: Line 398: Change form on: „the plants were watered thrice (the same for line 402: the leaves and (…) were harvested..; and for line 418… - check all text)

Author Response

Thank you for the constructive comments on my work. In the following, I will comment on the individual points (in blue):

Manuscript is interesting and quite well written. I have only some comments regarding the organization:

Thank you!

Tables and Figures should be placed close to the point where they were cited in the text to facilitate following the content. Figures should be cited first and then placed in the text (comment for Fig. 2).

Thank you for the hint, I have corrected this now.

Line 255: Fig. 3 should be cited before Fig. 4

Done. I changed the order and the former Fig. 3 is now Fig. 4 and vice versa.

Line 373: Why was reference [50] placed here?

I used plant seeds from the same plants as in reference [50] and the reference describes the seed collection in detail. I have now clarified this in the text.

The scientific text are rather written in passive voice: Line 398: Change form on: „the plants were watered thrice (the same for line 402: the leaves and (…) were harvested..; and for line 418… - check all text)

Done.

Reviewer 3 Report

1- In section "2.2 Ecological relevance of terpenoids in E. cicutarium" nothing was said about your results and the section looks like an introduction or discussion (without comparing with your findings) .

2- In subtitles species name are not in italic.

3- I think (suggest) it would be usefull if you mark which compounds were identified specifically in Bavaria (BY) and California origin plants inTable 1.

Good luck.

Author Response

Dear reviewer,

thank you for the constructive comments on my work. In the following, I will comment on the individual points (in blue):

1- In section "2.2 Ecological relevance of terpenoids in E. cicutarium" nothing was said about your results and the section looks like an introduction or discussion (without comparing with your findings) .

Yes, I agree and added a sentence to section 2.2. In this manuscript, the results and discussion chapters are merged.

2- In subtitles species name are not in italic.

To my knowledge, when a title or sentence is italicized, a word that normally would be italicized inside this running text should appear in roman type (referred to as ‘reverse italics’.) At least this is stated in the Editorial Style Guide, Purchase College, State University of New York.

3- I think (suggest) it would be usefull if you mark which compounds were identified specifically in Bavaria (BY) and California origin plants inTable 1.

I agree and added the information.

Good luck.

Thank you!

Round 2

Reviewer 1 Report

Dear Author,

I am glad to see your efforts to improve your manuscript. Although you didn't follow my advice and the advice of Reviewer 2 to rewrite the text in the passive voice, it is now easier to follow.  

I have only one request, to move Figure 1 after paragraph L244-257 where it is mentioned for the first time. 

Good luck

Author Response

Dear Reviewer,

thank you very much for your comments in the 2nd revision round. I did change the methods section now to passive voice. The Fig. 1 was first mentioned in line 112,
therefore I did not move it further down in the text and I hope that you agree to this. 

Best regards and thank you.